# The GluTR-binding protein is the heme-binding factor for feedback control of glutamyl-tRNA reductase

Andreas S Richter*, Claudia Banse, Bernhard Grimm*

Institute of Biology/Plant Physiology, Humboldt-Universität zu Berlin, Berlin, Germany

**Abstract** Synthesis of 5-aminolevulinic acid (ALA) is the rate-limiting step in tetrapyrrole biosynthesis in land plants. In photosynthetic eukaryotes and many bacteria, glutamyl-tRNA reductase (GluTR) is the most tightly controlled enzyme upstream of ALA. Higher plants possess two GluTR isoforms: GluTR1 is predominantly expressed in green tissue, and GluTR2 is constitutively expressed in all organs. Although proposed long time ago, the molecular mechanism of heme-dependent inhibition of GluTR *in planta* has remained elusive. Here, we report that accumulation of heme, induced by feeding with ALA, stimulates Clp-protease-dependent degradation of *Arabidopsis* GluTR1. We demonstrate that binding of heme to the GluTR-binding protein (GBP) inhibits interaction of GBP with the N-terminal regulatory domain of GluTR1, thus making it accessible to the Clp protease. The results presented uncover a functional link between heme content and the post-translational control of GluTR stability, which helps to ensure adequate availability of chlorophyll and heme.
DOI: https://doi.org/10.7554/eLife.46300.001

*For correspondence:
andreas.richter@hu-berlin.de
(ASR);
bernhard.grimm@hu-berlin.de
(BG)

Competing interests: The authors declare that no competing interests exist.

## Introduction

Tetrapyrrole biosynthesis (TBS) is one of the vital biochemical pathways found in organisms. Besides providing heme and its breakdown products, which are essential constituents of almost all organisms, the pathway supplies the chlorophyll (Chl) crucial for photosynthesis in cyanobacteria, algae, and plants (*Tanaka and Tanaka, 2007*). In plants, the plastid-localized TBS is initiated by the reduction of a tRNA-coupled glutamate to glutamate-1-semialdehyde (GSA) by glutamyl-tRNA reductase (GluTR). The unstable GSA is then converted into 5-aminolevulinic acid (ALA) by GSA aminotransferase (GSAT). ALA is the universal precursor of all tetrapyrroles, and its synthesis is considered to be rate-limiting for TBS as a whole. Eight molecules of ALA are converted into the macrocyclic protoporphyrin IX (ProtoIX) in subsequent reactions. Thereafter, ProtoIX is directed into either the iron or magnesium branch for the synthesis of heme or Chl, respectively (*Tanaka and Tanaka, 2007*). For heme synthesis in plants, two related ferrochelatases (FC1 and FC2) are available, which are characterized by distinct and partially overlapping functions (*Espinas et al., 2016*; *Fan et al., 2019*; *Hey et al., 2016*; *Scharfenberg et al., 2015*; *Woodson et al., 2011*). Synthesis of Chl is initiated by the insertion of $Mg^{2+}$ into ProtoIX, catalyzed by the Mg-chelatase (MgCh) complex. Two additional steps convert the catalytic product Mg protoporphyrin IX (MgP) into protochlorophyllide (Pchlide) which, in angiosperms, is reduced to chlorophyllide (Chlide) by the light-dependent protochlorophyllide oxidoreductase (POR). Chlide is converted to Chl *a* and *b* by enzymes of the Chl cycle (*Tanaka and Tanaka, 2007*).

Various regulatory mechanisms of TBS have been reported, which modify the activities and amounts of the TBS enzymes, including GluTR (*Brzezowski et al., 2015*; *Tanaka et al., 2011*). Thus, transcriptional and post-translational regulatory mechanisms ensure the provision of adequate levels

of active GluTR to ensure balanced ALA synthesis rates. Post-translational modifications of GluTR include redox regulation (*Richter et al., 2013*; *Wittmann et al., 2018*), dark- and low-light-dependent suppression of its activity by interaction with FLUORESCENT IN BLUE LIGHT (FLU) (*Hou et al., 2019*; *Meskauskiene and Apel, 2002*; *Meskauskiene et al., 2001*), folding by the chaperone chloroplast signal recognition particle 43 (*Wang et al., 2018*), as well as the conditional proteolysis of GluTR by the caseinolytic protease (Clp) and regulation of its stability by the GluTR-binding protein (GBP) (*Apitz et al., 2016*; *Czarnecki et al., 2011*; *Nishimura et al., 2015*).

The plastid-localized *Arabidopsis* Clp protease uses the selector proteins ClpS1 and ClpF1 and the chaperones ClpC1/2 and ClpD (*Nishimura and van Wijk, 2015*; *Nishimura et al., 2013*). These subunits are required for recognition and ATP-driven unfolding of the substrate proteins prior to the proteolytic cleavage (*Olinares et al., 2011*). Binding of the N-terminus of *Arabidopsis* GluTR1 to ClpS, ClpF, and ClpC induces its proteolysis (*Apitz et al., 2016*; *Nishimura et al., 2015*). Turnover of bacterial GluTR is analogously catalyzed by Clp and the Lon protease (*Wang et al., 1999a*). Plant GBP, which has no homolog in bacteria, competes with the Clp subunits for binding to the N-terminus of GluTR, and thus counteracts GluTR breakdown (*Apitz et al., 2016*). The GluTR site recognized by both GBP and Clp subunits includes a 30-amino-acid N-terminal peptide, which was previously designated as the 'heme-binding domain' (HBD). Deletion of the HBD was found to compromise heme-dependent inhibition of GluTR activity in vitro (*Vothknecht et al., 1998*) and Clp-dependent degradation of GluTR in plants (*Apitz et al., 2016*). Due to the negative impact of heme on GluTR activity, it was proposed that heme feedback-controls GluTR and, consequently, ALA synthesis rates (*Cornah et al., 2003*; *Terry and Kendrick, 1996*; *Terry and Kendrick, 1999*; *Vothknecht et al., 1998*).

We set out to identify factors that affect heme-dependent regulation of GluTR activity and stability. Here, we present experimental evidence for a direct role of heme in regulating the proteolysis of GluTR1 by the Clp protease. The results presented in this study reveal that the N-terminus of GluTR – a putative heme-binding domain (HBD) – is not involved in heme binding. The importance of heme for GluTR1 stability derives from its interaction with GBP. Based on these new findings, and in light of recent data that underline the regulatory impact of the N-terminal region of GluTR1 (*Wang et al., 2018*), we refer to the 'HBD' as the '<u>r</u>egulatory <u>d</u>omain' (RED) of GluTR1 in the following.

## Results

### Induced destabilization of GluTR upon ALA feeding

To explore the impact of TBS metabolites on the stability of GluTR and several other plastid-localized proteins, *Arabidopsis* wild-type plants (WT, Col-0) were transferred to buffer supplemented or not with 1 mM ALA. The water-soluble ALA is rapidly taken up through the leaf surface and metabolized in plastids, leading to substantially increased steady-state levels of Pchlide (*Figure 1*). Side-effects of reactive oxygen species (ROS) arising from accumulating tetrapyrroles were avoided by dark incubation of detached leaves. Photoperiodic interference owing to diurnal variations in gene expression was excluded by first growing the plants in continuous light. Western blot analysis was used to monitor patterns of protein accumulation in response to the treatments.

First, we found that a group of proteins including GluTR2, GSAT, MgCh subunit I (CHLI), GENOMES UNCOUPLED 4 (GUN4), the CHL27 subunit of the MgPMME oxidative cyclase (CHL27, *Pontier et al., 2007*), the product of hypothetical open reading frame 54 [YCF54/LCAA, a subunit of Mg-protoporphyrin IX monomethylester (MgPMME) oxidative cyclase], ferrochelatase (FC) 2, Clp protease C subunit, and subunits of the photosynthetic protein complexes, such as LHCA1/B1, PetB and PsaL, showed little or no variation in the course of a 24 hr dark incubation (*Figure 1A*). GBP, Mg-protoporphyrin IX (MgP) methyltransferase (CHLM), and ClpS were detected in reduced amounts in both control and ALA-fed leaves. Hence, these effects are not a direct consequence of ALA feeding (*Figure 1A*). In contrast, we identified two proteins with differential accumulation patterns. Accumulation of FC1 was strongly induced, and GluTR1 content decreased in control leaves in the dark, but these effects became markedly more pronounced in the presence of supplied ALA. In comparison to the light-grown seedlings at t(0), GluTR1 amounts were slightly reduced after 4 hr and reduced by 40% after 24 hr of darkness without ALA supply. But even short term (4 hr) ALA feeding triggered a drastic loss of GluTR1 (76% reduction relative to t(0)), while the

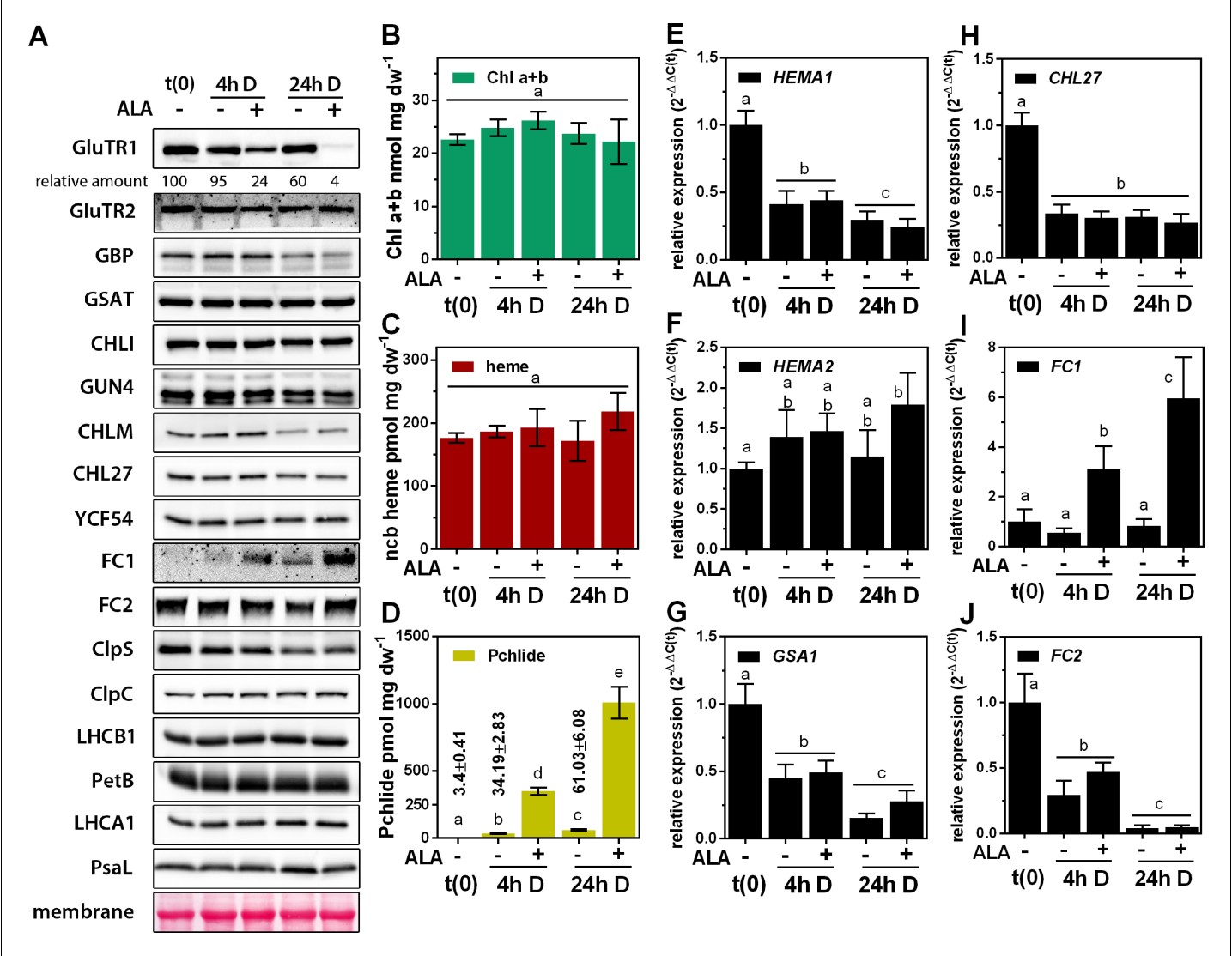

**Figure 1.** Post-translational destabilization of GluTR1 upon feeding with 5-aminolevulinic acid (ALA) in the dark. Fourteen-day-old, light-grown *Arabidopsis* Col-0 wild-type plants were incubated in buffer without (-) or with 1 mM ALA (+) in darkness (D) for the indicated times. Samples were harvested prior to treatment (t0) and at 4 hr and 24 hr after the onset of treatment. (**A**) Western blot analysis of proteins involved in tetrapyrrole biosynthesis (GluTR1/2, GBP, CHLI, GUN4, CHLM, CHL27, YCF54, FC1, and FC2), Clp protease (ClpS and C) and components of the photosynthetic electron transfer chain (LHCA1, LHCB1, PetB, PsaL). t(0) = sample harvested from light-grown seedlings prior to ALA treatment. Signal intensities for GluTR1 relative to t(0) are shown at the top. (**B – D**) Levels of chlorophylls (Chl) a and b (**B**), non-covalently bound (ncb) heme (**C**) and protochlorophyllide (Pchlide) (**D**) found in control and ALA-treated seedlings. Data are given as means ± sd (n = 4). (**E – J**) qRT-PCR analysis of *HEMA1* (coding for glutamyl-tRNA reductase 1), *HEMA2* (GluTR2), *GSA1* (glutamate-1-semialdehyde 2,1-aminomutase 1), *CHL27* (a subunit of aerobic cyclase) and *FC1/2* (ferrochelatase 1 and 2) mRNAs. Data are given as means ± sd (n = 4). The lowercase letters in panels (**B – J**) indicate statistical groups determined by Student's t-test (p<0.05).

DOI: https://doi.org/10.7554/eLife.46300.002

enzyme was barely detectable after 24 hr of ALA treatment (96% reduction relative to t(0), *Figure 1A*). This strong ALA-induced proteolysis was exclusively observed for GluTR1. Furthermore, Chl *a* and *b* (Chl a + b, *Figure 1B*) and non-covalently bound (ncb) heme (*Figure 1C*) content were not changed in ALA- and mock-treated leaves during 24 hr darkness compared to t(0). In contrast, the Pchlide levels increased by 10- and 20-fold in control and approximately 100- and 300-fold in ALA-fed leaves after 4 hr and 24 hr of dark treatment, respectively, in comparison to the steady-state levels detected in the light-grown seedlings (*Figure 1D*).

To dissect the molecular mechanism behind the remarkable decrease in GluTR1 content after ALA feeding, qRT-PCR analyses of genes encoding TBS enzymes were performed. Irrespective of ALA supplementation, levels of *HEMA1* (encoding GluTR1), *CHL27*, *GSA1* (encoding GSAT) and *FC2* mRNAs were decreased by at least 60% and up to 95% after 4 hr and 24 hr of dark incubation, respectively (*Figure 1E,H,G,J*). The *HEMA2* transcript content remained high during dark incubation and was slightly induced by 24 hr of ALA feeding (*Figure 1F*), while *FC1* mRNA content increased steadily upon incubation of leaves with ALA (*Figure 1I*). Amounts of GluTR2 and FC1 correlated with the corresponding mRNA levels detected throughout the 24 hr treatment (*Figure 1A*). However, *HEMA1* transcript levels were reduced during dark incubation regardless of the ALA supply, indicating that excess ALA had no specific effect on the expression of *HEMA1* (*Figure 1E*). In summary, it can be concluded that the decreased GluTR1 content observed upon ALA feeding is not a consequence of altered *HEMA1* transcription, but is instead attributable to post-translational destabilization of the protein.

## ALA-induced destabilization of GluTR involves the Clp protease

For the analysis of the impact of the Clp protease on ALA-dependent GluTR degradation, *Arabidopsis* WT, *clpc1-1,* and *clpr2-1* leaves were incubated in buffer without or with ALA for 4 hr in darkness (*Figure 2*). While the GluTR1 content was reduced by 74% in ALA-fed WT leaves compared to the control conditions without ALA, the ALA-incubated *clp* leaves retained 50% of the starting level of GluTR1 after 4 hr (*Figure 2A*). Both mutants accumulated WT-like amounts of Pchlide and ncb heme, indicating that lower Clp activity does not interfere with ALA uptake or metabolic conversion during the incubation period (*Figure 2B and C*). While GluTR1 was almost undetectable in ALA-fed WT leaves after 24 hr, ALA-treated *clpc1-1* and *clpr2-1* leaves still contained 22–25% of the GluTR1 content detected in untreated samples (*Figure 2D*). Thus, the altered Clp activity is correlated with delayed ALA-induced degradation of GluTR1. Additionally, diminished Clp protease activity is accompanied by more considerable GluTR1 accumulation under otherwise standard conditions (*Figure 2E*).

In agreement with previous results (*Apitz et al., 2016*), the lack of GBP in the *gbp-2* mutant is associated with a 40% reduction in GluTR1 content compared to the WT sample after 24 hr of dark incubation (*Figure 3B*). The 24 hr ALA treatment provoked severe degradation of GluTR1 in WT, and essentially complete loss of the protein in *gbp-2* leaves (*Figure 3B*). The importance of its N-terminal RED for the ALA-induced degradation of GluTR was examined with two *hema1* mutant lines expressing either *pHEMA1::HEMA1* or *pHEMA1::HEMA1(ΔRED)* encoding the complete and a truncated GluTR1, respectively. Both transgenic lines contained similar amounts of GluTR1 to the WT control after 24 hr of incubation in ALA-free buffer (*Figure 3B*). Strikingly, however, and in contrast to WT and *hema1/HEMA1* leaves, truncated GluTR1 was not degraded in the *hema1/HEMA1(ΔRED)* line upon ALA feeding (*Figure 3B*). The amounts of Pchlide and ncb heme detectable after 4 hr of incubation were comparable across all genotypes (*Figure 4C,D*).

## Identification of the TBS metabolite responsible for triggering the degradation of GluTR1

Whether ALA itself or a tetrapyrrolic intermediate or end-product is responsible for GluTR1 degradation was examined in ALA-feeding experiments performed in the presence of inhibitors of specific steps in TBS. We made use of a photoautotrophic *Arabidopsis* cell culture (PA) (*Hampp et al., 2012*), which was simultaneously exposed to ALA and inhibitors of TBS (*Figure 4A*). Acifluorfen (ACI) inhibits PPOX and prevents the accumulation of Pchlide upon ALA feeding (*Figure 4B*), while 2,2'-dipyridyl (DP) restricts Pchlide and heme formation by inhibiting the MgPMME cyclase and FC, respectively (*Figure 4B*). Apart from a slight reduction after 24 hr of DP and ALA-treatment, ncb heme levels remained stable in the PA cells during the incubation period (*Figure 4C*). Intriguingly, however, both ACI and DP prevented the ALA-induced degradation of GluTR1 in the PA cultures (*Figure 4D*).

To investigate whether a metabolite involved in the heme or Chl branch serves as the ultimate stimulus for GluTR1 degradation, leaves of the Chl-free *Arabidopsis gun5-2* mutant (*CHLH* null mutant, *Huang and Li, 2009*) were incubated with ALA (*Figure 4—figure supplement 1*). After 4 hr of dark incubation, GluTR1 content was diminished in *gun5-2* leaves in the absence of ALA, and

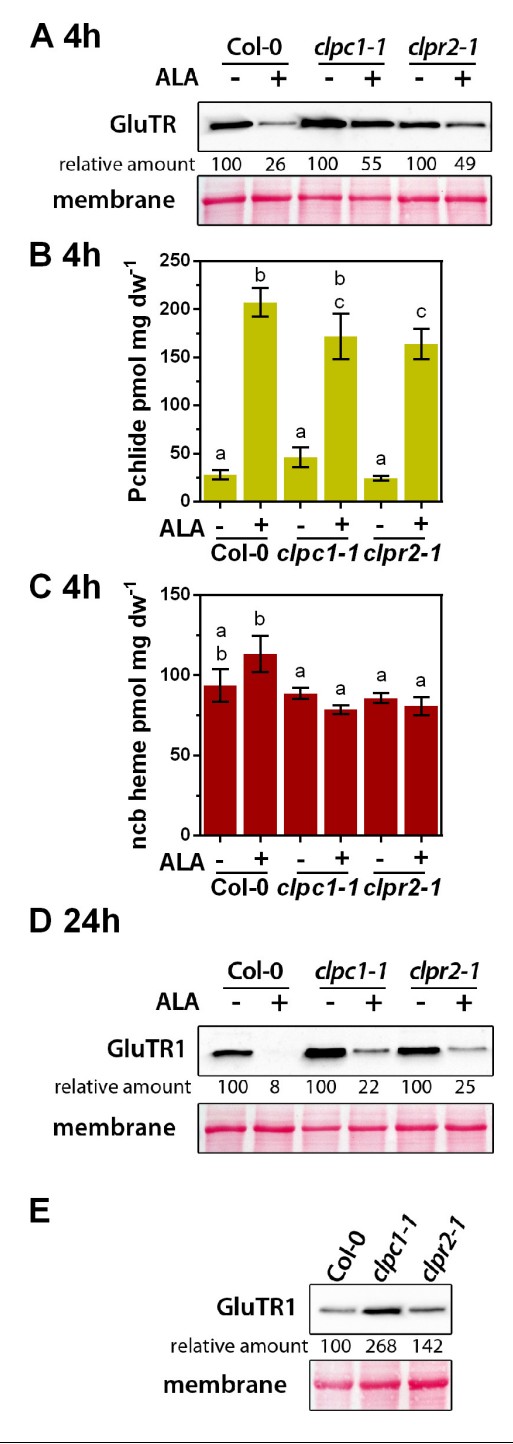

**Figure 2.** ALA-induced proteolysis of GluTR1 is attenuated in *clpc1-1* and *clpr2-1* mutants. (**A**) Western blot analysis of 14-day-old GluTR1 in Col-0, *clpc1-1* and *clpr2-1* leaves incubated in buffer without (-) or with (+) 1 mM ALA for 4 hr in darkness. Signal intensities for GluTR1 relative to Col-0 without ALA treatment are shown. (**B, C**) Pchlide (**B**) and non-covalently bound (ncb) heme (**C**) contents of genotypes analyzed in (**A**) after 4 hr of treatment. Data are given as means ± sd
*Figure 2 continued on next page*

even more strongly in the presence of ALA (*Figure 4E*). Proteolysis led to almost undetectable levels of GluTR1 in *gun5-2* leaves after 24 hr of dark incubation (±ALA), pointing to the heme branch as the source of the degradation signal of GluTR1. This suggestion was supported by the finding that feeding with hemin, but none of other tested tetrapyrroles triggered GluTR1 instability in isolated *Arabidopsis* chloroplasts (*Figure 4F*).

## Evidence for heme-dependent regulation of GluTR stability

To unequivocally verify that heme is involved in GluTR1 degradation, we employed *fc2-2* leaves (lacking the predominant FC2 isoform) for ALA-feeding experiments. GluTR was hypothesized to be more stable when less heme is available owing to impaired heme synthesis. Like the *clp* mutants, *fc2-2* accumulated more GluTR1 relative to WT levels at t(0)) under standard growth conditions (*Figure 5B*). While 4 hr of ALA-feeding induced GluTR1 proteolysis in WT, the starting content of GluTR1 remained stable in *fc2-2* (*Figure 5B*). In contrast to WT, GluTR1 proteolysis was clearly altered in *fc2-2* relative to t(0) after 4 hr and 24 hr ± ALA (*Figure 5B*). However, it is worth mentioning that GluTR1 levels in *fc2-2* leaves were lower after 24 hr of ALA treatment compared to t(0), which was not due to diminished Clp content (*Figure 5B*) nor of lower *HEMA1* expression (*Figure 5C*). Levels of, for example, *HEMA1* and *GSA1* transcripts (*Figure 5D*) in both control and ALA-treated *fc2-2* leaves were similar to those in WT analyzed under identical conditions (*Figure 1*). However, overaccumulation of FC1 was observed in detached *fc2-2* leaves (*Figure 5B*). The ALA-induced accumulation of ProtoIX (*Figure 5E*) and Pchlide (*Figure 5F*) was similar in dark-incubated WT and *fc2-2* leaves. In comparison to WT without ALA, *fc2-2* accumulated ProtoIX during light exposure and Pchlide in darkness (*Figure 5E and F*) leading to a significant decrease of ncb heme at t(0), but no further decline during dark incubation (both with -ALA and +ALA) (*Figure 5G*).

The previously introduced term 'free heme' refers to heme synthesized by FCs, which is available for further metabolic conversion (*Espinas et al., 2012*). While dark incubation of WT leaves in ALA-free buffer did not modify the level of intracellular free heme, supplementation with ALA significantly increased the content of free heme in the WT after 4 hr and 24 hr, respectively (*Figure 5H*). In contrast, the overall free heme content was significantly reduced in the

*Figure 2 continued*

(n = 3). Letters indicate statistical groups determined by Student's t-test (p<0.05). (D) Western analysis of GluTR1 in Col-0, *clpc1-1* and *clpr2-1* incubated in buffer without (-) or with (+) 1 mM ALA for 24 hr in darkness. Signal intensities for GluTR1 in protein extracts of the genotypes treated without (-ALA = 100%) and with ALA (+ALA) are shown. (E) GluTR1 content of Col-0, *clpc1-1* and *clpr2-1* seedlings grown under short-day conditions. Samples were harvested 2 hr after the onset of light. Signal intensities for GluTR1 are shown relative to Col-0.
DOI: https://doi.org/10.7554/eLife.46300.003

*fc2-2* leaves, and FC2 deficiency prevented accumulation of free heme upon ALA feeding (*Figure 5H*). For comparison, an *FC1* knockdown mutant (*FC1* knockout is embryo-lethal) showed WT-like GluTR1 content upon ALA supplementation in darkness (*Figure 5—figure supplement 1*). In summary, genetic impairment of heme biosynthesis caused by the lack of plastid-localized FC2 compromises ALA-induced GluTR1 degradation.

## Binding of heme to GBP interferes with GluTR1 interaction

The potential impact of heme on GluTR1 and GBP was explored by analyzing heme-binding to recombinant variants of these two proteins in vitro. The proteins were separately incubated with hemin-coupled agarose (HA) or control agarose matrix (*Figure 6*). Both the GST- and HIS-tagged GBP proteins were able to bind to HA, but not to unmodified agarose (*Figure 6A*), while none of the tagged GluTR1 variants bound to HA (*Figure 6B*). These results confirm that GBP binds specifically to the HA. Analysis of in vitro pull-down assays with recombinant 6xHIS-GluTR and GST-GBP in the absence (*Figure 7A*) or presence of hemin (*Figure 7B*) revealed GST-GBP binding to 6xHIS-GluTR1 in the absence of heme, while added hemin abolished this interaction (*Figure 7*). This effect of heme on GluTR1-GBP complex formation was verified using the microscale thermophoresis (MST) approach (*Figure 7C/D*). Increasing amounts of 6xHIS-GluTR1 or 6xHIS-GBP were titrated against fluorescently labeled 6xHIS-GluTR1 or 6xHIS-GBP. When GluTR1 was titrated against labeled GBP (*Figure 7C*), a Kd value of 65 nM was calculated. Likewise, the Kd value for the GBP-GluTR1-FLUO interaction was 113 nM (*Figure 7D*). Heat inactivation of GluTR1 abolished the interaction with GBP-FLUO (*Figure 7—figure supplement 1*). Finally, addition of 1 µM hemin markedly decreased the affinity between the two proteins by 7- to 14-fold (GBP-FLUO: 931 nM / GluTR1-FLUO: 806 nM) (*Figure 7C/D*). Together, these experiments demonstrate the heme-binding ability of GBP and the negative impact of heme on the interaction of GBP and GluTR1.

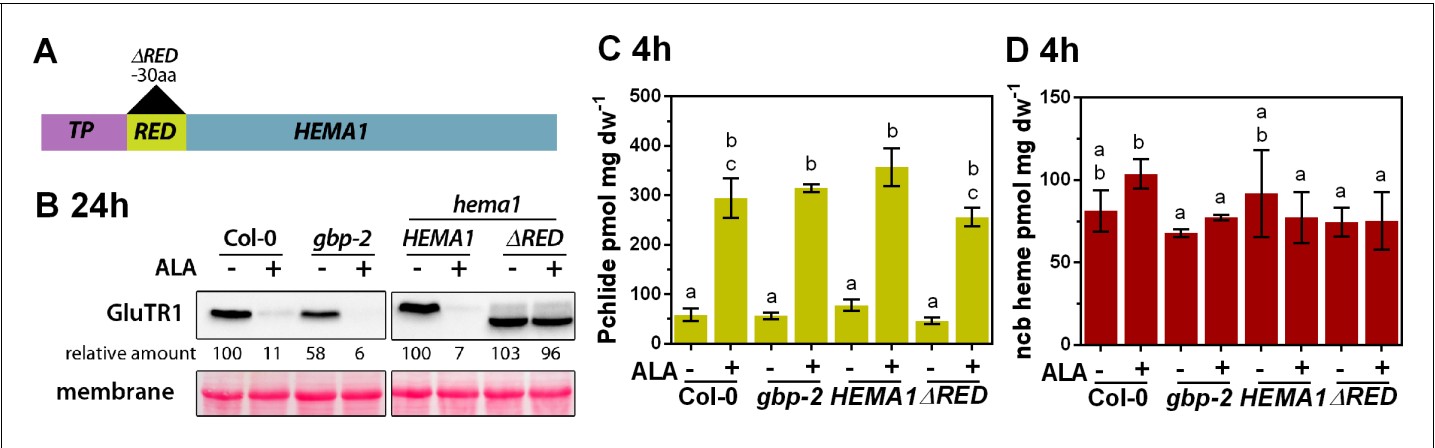

**Figure 3.** ALA-induced proteolysis depends on the N-terminal segment of GluTR1 and on GBP. (A) Schematic presentation of the constructs used to complement a *hema1* knockout mutant. In addition to the full-length coding sequence of *HEMA1*, a construct coding for a truncated version of GluTR1 (ΔRED), which lacks 30 amino acids (aa) of the regulatory domain, was used to complement the *hema1* mutant. Transgenes were under the control of the native *HEMA1* promoter. TP, transit peptide. (B) GluTR1 content of 14-day-old Col-0, *gbp-2*, *hema1/HEMA1*, and *hema1/ΔRED* leaves incubated in buffer without (-) or with (+) 1 mM ALA for 24 hr in darkness. Signal intensities for GluTR1 are shown relative to the untreated WT (left) or the *hema1/HEMA1* line (right). (C, D) Pchlide (C) and ncb heme content (D) of genotypes analyzed in (B) after 4 hr of treatment. Data are given as mean ± sd (n = 3). Letters indicate statistical groups determined by Student's t-test (p<0.05).
DOI: https://doi.org/10.7554/eLife.46300.004

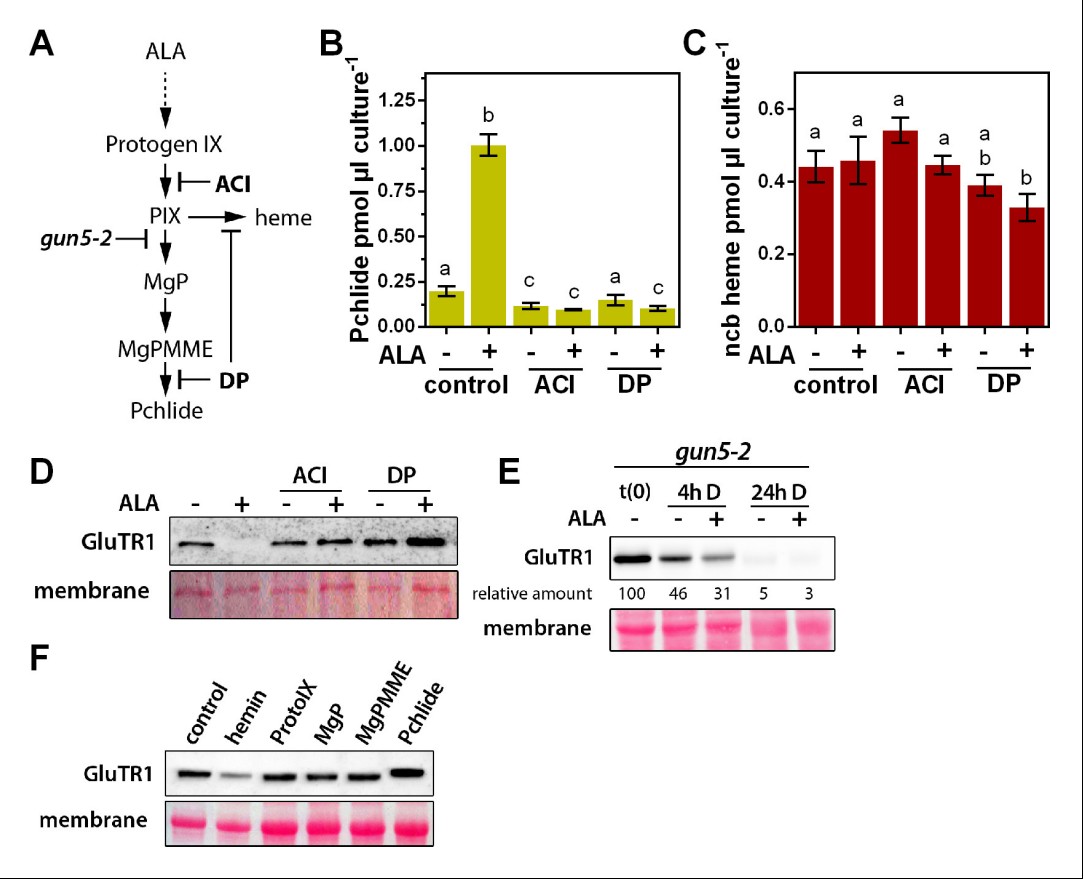

**Figure 4.** Identification of the stimulus responsible for triggering GluTR1 degradation. A photoautotrophic *Arabidopsis* cell culture was used, together with mutants for the tetrapyrrole biosynthesis pathway, and isolated chloroplasts. (A) Overview of the tetrapyrrole biosynthesis pathway. Acifluorfen (ACI) prevents the formation of protoporphyrin IX (PIX) by inhibiting protoporphyrinogen oxidase. The iron chelator 2,2'-dipyridyl (DP) inhibits the reactions catalyzed by the ferrochelatases (FC) and aerobic cyclase, respectively. Knockout of *GUN5* (*gun5-2*) prevents synthesis of Mg-protoporphyrin IX (MgP) and downstream intermediates. (B, C) Pchlide (B) and ncb heme (C) contents of photoautotrophically cultured *Arabidopsis* cells (PA) incubated in the presence (+) or absence of (-) 1 mM ALA. In addition to ALA, cells were treated with 100 µM ACI or 500 µM DP, respectively. PA cultures were pre-incubated with ACI or DP for 2 hr in darkness, before the buffer was exchanged for media supplemented with ACI, DP, and ALA. Treatment was performed for 16 hr in the dark. Data are given as mean ± sd (n = 4). Letters indicate statistical groups determined by Student's t-test (p<0.05). (D) Levels of GluTR1 in cell cultures incubated in the absence (- ALA) or presence (+ALA) of added ALA and with or without ACI or DP, respectively. (E) Levels of GluTR1 in leaves of *gun5-2* knockout mutants incubated without (-) or with (+) 1 mM ALA in the dark (D) for the indicated times. Signal intensities for GluTR1 relative to t(0) are shown. (F) *Arabidopsis* Col-0 chloroplasts were isolated and incubated for 6 hr in darkness in the presence of hemin and intermediates of the tetrapyrrole biosynthesis pathway. Pchlide was used at 1 µM and heme and the other porphyrins at 5 µM final concentration.
DOI: https://doi.org/10.7554/eLife.46300.005

The following figure supplement is available for figure 4:

**Figure supplement 1.** Phenotype (A) and genotyping (B) of Col-0 and the *gun5-2* mutant.
DOI: https://doi.org/10.7554/eLife.46300.006

## Discussion

The results presented here uncover a new regulatory link between the previously proposed heme-dependent control of ALA synthesis (*Cornah et al., 2003*) and the proteolysis of GluTR (*Apitz et al., 2016*) and, thus, can answer the fundamental and longstanding question of the molecular function of heme in the control of ALA synthesis in plants.

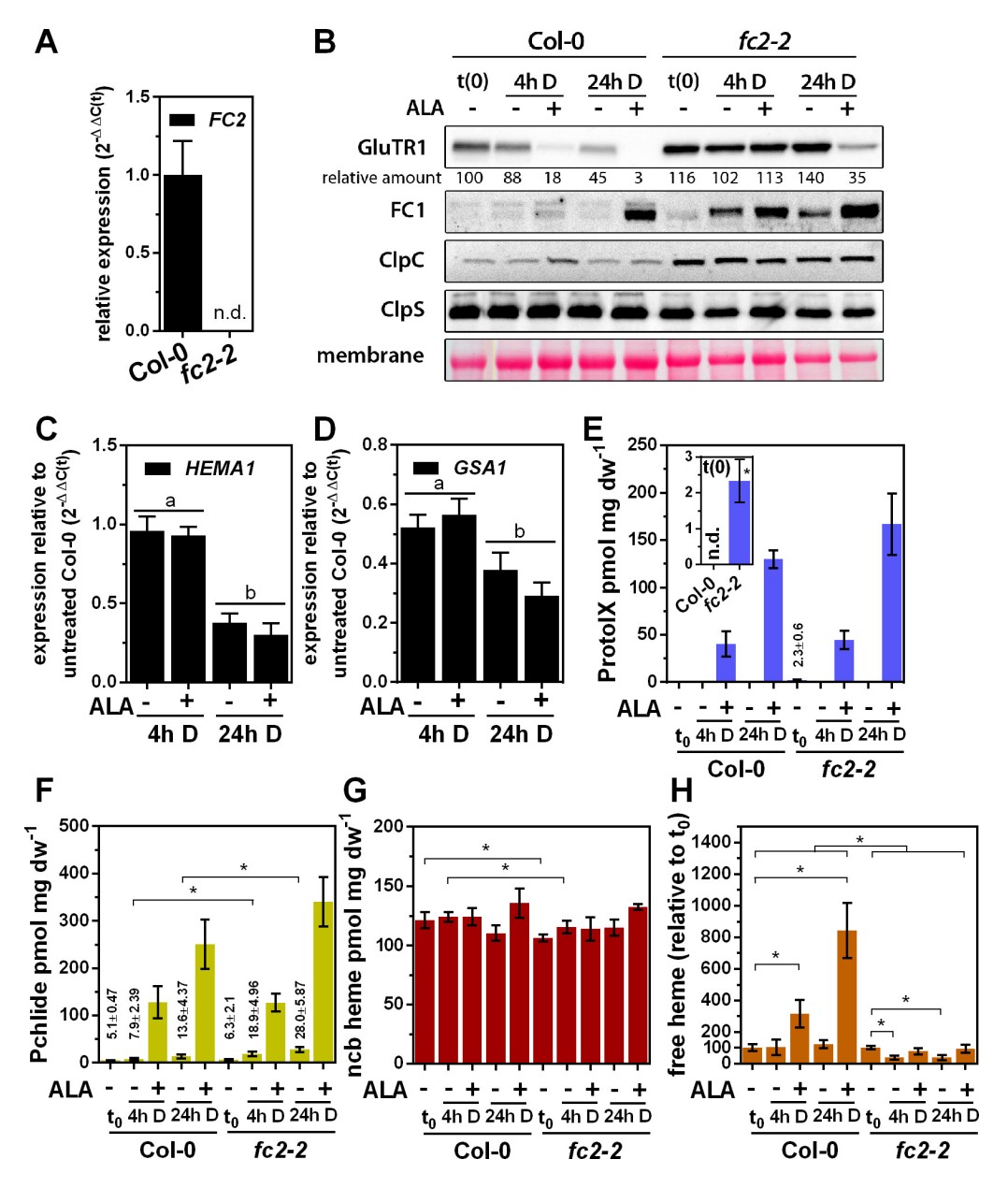

**Figure 5.** Altered ALA-induced proteolysis of GluTR1 in *FC2* knockout mutants. (**A**) Levels of *FC2* mRNA in light-grown Col-0 and *fc2-2* plants, analyzed by qRT-PCR. Data are given as mean ± sd (n = 3). n.d., not detectable (**B**) Levels of GluTR1, FC1, ClpC and ClpS in leaves of Col-0 and *fc2-2* plants incubated without (-) or with (+) 1 mM ALA in darkness (D) for the indicated times. Signal intensities for GluTR1 relative to Col-0 seedlings harvested before the beginning of the experiment (Col-0 t(0)) are shown (D, dark). (**C**) and (**D**) Levels of *HEMA1* (**C**) and *GSA1* (**D**) mRNAs in *fc2-2* mutants plotted relative to untreated Col-0 seedlings harvested before start of the experiment. Data are given as mean ± sd (n = 4). Letters indicate statistical groups determined by Student's t-test (p<0.05). (**E**) Protoporphyrin IX (ProtoIX), (**F**) Pchlide, (**G**) ncb heme and (**H**) free heme contents of Col-0 and *fc2-2* leaves treated as in (**B**). The inset in E shows the steady-state level of ProtoIX in untreated seedlings before ALA treatment (t0). Data are given as mean ± sd (n = 4). Asterisks denote statistically significant changes between samples with p<0.05 (Student's t-test). Statistical analysis was performed before the calculation of ratios. n.d., not detectable. 21-day-old, light-grown plants were analyzed.

DOI: https://doi.org/10.7554/eLife.46300.007

The following figure supplement is available for figure 5:

**Figure supplement 1.** ALA-induced proteolysis of GluTR1 in *FC1* knockdown mutants.

*Figure 5 continued on next page*

*Figure 5 continued*

DOI: https://doi.org/10.7554/eLife.46300.008

## Excess of ALA stimulates degradation of GluTR by Clp

GluTR1 is a well-established substrate for the Clp protease in plants (*Apitz et al., 2016*; *Nishimura et al., 2015*) and bacteria (*Wang et al., 1999a*), but the factors and signals that regulate the stability of GluTR have been elusive. In the presence of excess ALA, GluTR1 is more stable in the functional knockdown mutants *clpc1-1* and *clpr2-1* (*Kim et al., 2009*; *Nishimura and van Wijk, 2015*) than it is in WT (*Figure 2*). The GluTR1 N-terminal region, including the RED, is recognized by the corresponding Clp subunits (*Apitz et al., 2016*; *Nishimura et al., 2015*). Consequently, deletion of the RED renders GluTR1 insensitive to ALA-induced Clp degradation in darkness and prevents proteolysis of ΔRED-GluTR (*Figure 3B*). In contrast, knockout of *GBP* (which otherwise counteracts proteolysis) resulted in faster breakdown of GluTR than that seen in WT (*Figure 3B*). The rapid degradation of GluTR1 in leaves supplemented with ALA (*Figure 1A*), which is not correlated with declined transcription of *HEMA1* (*Figure 1E*), highlights a posttranslational mechanism that specifically controls GluTR1 availability for TBS by excess ALA.

## Heme initiates GluTR1 proteolysis

Two earlier findings prompted us to hypothesize that TBS metabolites are likely involved in ALA-promoted GluTR1 degradation. Firstly, accumulation of Pchlide has been reported to induce the FLU-dependent repression of GluTR activity in darkness (*Meskauskiene et al., 2001*; *Richter et al., 2010*). Thus, dark supplementation with ALA, which would drastically increase levels of TBS intermediates, in particular Pchlide, could also function as a feedback signal for the immediate breakdown of GluTR1. Secondly, heme-dependent feedback regulation of GluTR was suggested in previous studies, and an N-terminally truncated recombinant GluTR1 was shown to be insensitive to it (*Terry and Kendrick, 1996*; *Terry and Kendrick, 1999*; *Vothknecht et al., 1998*).

Multiple lines of evidence support heme-mediated control of GluTR amounts in plants. 1. Inhibition of Pchlide and heme accumulation at the levels of PPOX, FC and oxidative cyclase, respectively, prevents ALA-induced GluTR1 proteolysis in a photoautotrophic cell culture (*Figure 4D*). By using two chemically unrelated inhibitors of the TBS pathway, we ruled out unspecific effects of these compounds on the Clp protease and excluded the possibility that ALA or an intermediate upstream of ProtoIX serves as the signal for GluTR1 degradation. 2. Supplementation of purified *Arabidopsis* chloroplasts with various tetrapyrroles identified heme as the stimulatory factor for GluTR1 degradation (*Figure 4F*). 3. The ALA-fed *gun5-2* knockout mutant exhibits ALA-induced GluTR1 degradation, revealing that the process is not triggered by metabolites of the Chl-synthesizing branch (*Figure 4E*). 4. The final proof for the feedback regulatory function of heme was obtained when ALA-fed *fc2-2* leaves were found to retain GluTR1 in the darkness (*Figure 5B*). We also

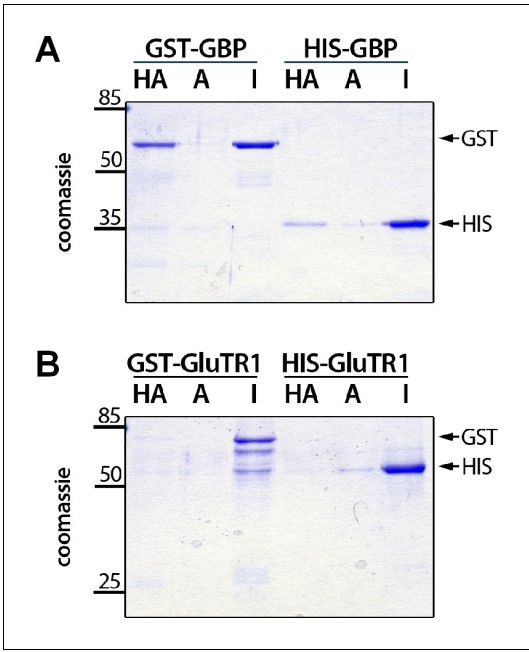

**Figure 6.** Binding of hemin to GBP. Recombinant GST- and 6xHIS-tagged GBP (**A**) and GluTR1 (**B**), respectively, were incubated with hemin-agarose (HA) or unmodified agarose (A), washed, eluted and subjected to electrophoresis on a 12% SDS-polyacrylamide gel. One microgram of the input was loaded as a reference (I). Note that only GBP binds to the HA. Arrows on the right mark the position of the GST and 6xHIS-tagged proteins.

DOI: https://doi.org/10.7554/eLife.46300.009

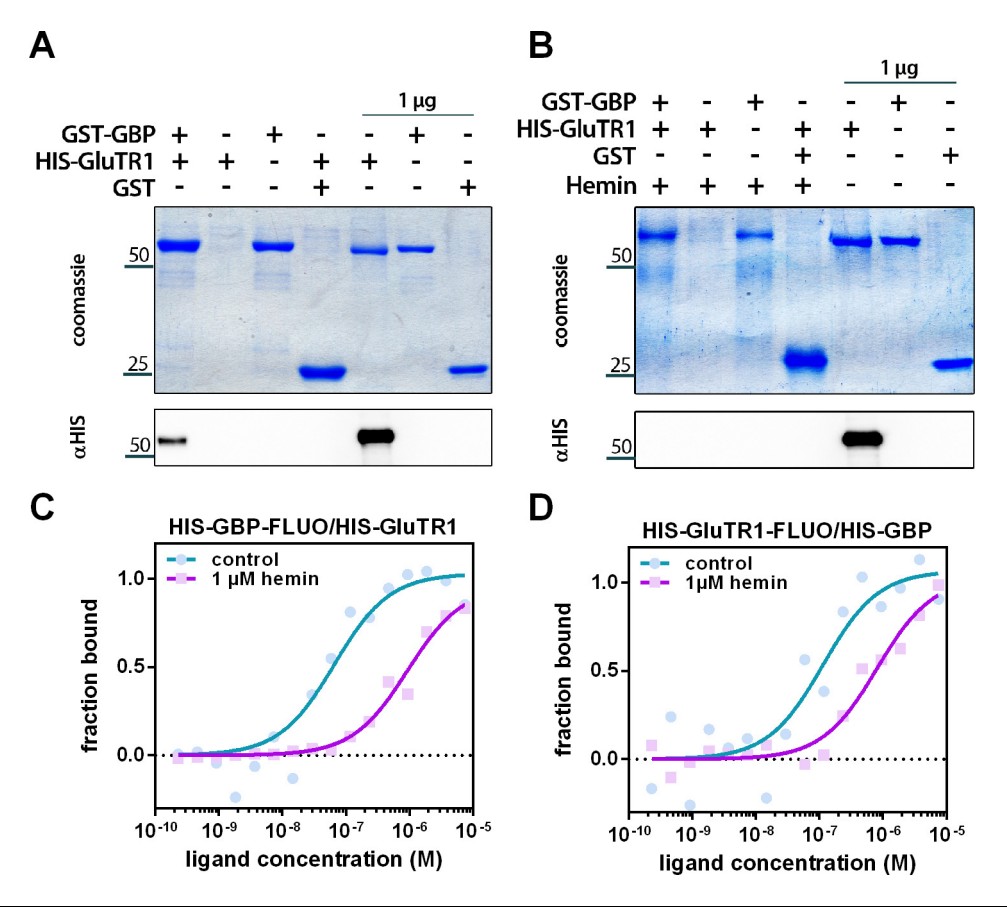

**Figure 7.** Hemin interferes with the binding of GBP to GluTR1. (**A**) Pulldown of 6xHIS-GluTR1 by GST-GBP. Both proteins were incubated in the presence of Glutathione-Sepharose (GS), washed, eluted using reduced glutathione and separated on a 12% SDS-polyacrylamide gel. Eluates of reactions containing GST-GBP alone or GST and GluTR1 as well as 1 µg of the input (no GS incubation) were loaded as controls. Note that 6xHIS-GluTR1 and GST-GBP show almost identical migration behavior. Successful pulldown of 6xHIS-GluTR1 by GST-GBP was confirmed using a HIS-tag-specific antibody (αHIS) after western blotting. (**B**) Same experiment as in (**A**) but carried out in the presence of 500 µM hemin. Note that hemin prevents pulldown of 6xHIS-GluTR1 by GST-BP. (**C**) Microscale thermophoresis (MST) of the fluorophore-labeled (FLUO) 6xHIS-GBP titrated against increasing amounts of 6xHIS-GluTR1. The affinity of 6xHIS-GBP-FLUO for 6xHIS-GluTR1 (Kd = 65 nM) decreases upon addition of 1 µM hemin (Kd = 931 nM). n = 2 independent experiments. (**D**) MST of fluorophore-labeled 6xHIS-GluTR1-FLUO titrated against increasing amounts of 6xHIS-GBP. The affinity of 6xHIS-GluTR1-FLUO for 6xHIS-GBP (Kd = 113 nM) decreases upon addition of 1 µM hemin (Kd = 806 nM). n = 2 independent experiments.
DOI: https://doi.org/10.7554/eLife.46300.010

The following figure supplement is available for figure 7:

**Figure supplement 1.** Control for MST measurement.
DOI: https://doi.org/10.7554/eLife.46300.011

observed significantly different contents of 'free' heme in ALA-fed WT and *fc2-2* leaves (*Figure 5H*). This latter result reflects the expected phenotype of deficient heme synthesis in the chloroplast, where FC2 is the dominant isoform. The increased FC1 content (*Figure 5B*) only partially compensated for the deficit of FC2 activity and did not lead to adequate heme synthesis in plastids. The specific role of FC2-derived heme for the control of ALA synthesis is further emphasized by the WT-like stability of GluTR1 in *FC1* knockdown mutants (*Figure 5—figure supplement 1*). Therefore, it is proposed that heme synthesized by FC2 is allocated for GBP-binding, while FC1 provides heme for other cellular compartments, as it has been proposed previously (*Espinas et al., 2016*; *Scharfenberg et al., 2015*). In this context, although measurements of 'free' heme (*Espinas et al.,*

*2012*) were used to show reduced FC activities in *fc2-2* compared to WT, the heme responsible for GluTR1 breakdown is most likely bound to a (regulatory) protein and belongs to the ncb fraction.

In conclusion, lack of heme synthesis by FC2 correlates with greater stability of GluTR1 during short-term ALA feeding in the darkness. Thus, the involvement of heme in the regulation of ALA synthesis constitutes one among the many post-translational mechanisms that control GluTR. GluTR1 degradation was indeed also observed in *fc2-2* after 24 hr of incubation with ALA, but was significantly less pronounced than in the WT (*Figure 5B*), indicating the existence of an additional proteolysis pathway that is independent of FC2-synthesized heme. Alternatively, the elevated FC1 content seen in *fc2-2* (*Figure 5B*) could also contribute to the synthesis of regulatory heme.

## Model: Binding of heme by GBP makes GluTR1 available for Clp-dependent proteolysis

Previous studies revealed a structural homology between GBP and the bacterial heme oxygenase HugZ, which converts heme into biliverdin (*Hu et al., 2011*; *Zhao et al., 2014*). Although the order of domains differs in GBP from that of HugZ and its mode of heme-binding has not been analyzed, co-filtration and isothermal titration calorimetry (ITC) have demonstrated binding of heme to GBP (*Zhao et al., 2014*). The hemin-agarose-based binding assays described here unequivocally validate GBP as a heme-binding protein (*Figure 6A*). The inability of GluTR1 to bind hemin in this assay (*Figure 6B*) agrees with the absence of bound heme in GluTR1 crystal structures (*Fang et al., 2016*; *Zhang et al., 2015*; *Zhao et al., 2014*).

The regulatory model shown in *Figure 8* depicts binding of heme-free GBP to GluTR1. Thus, we propose that GBP can bind either to GluTR or to heme, and as long as the GBP is bound to GluTR1, the recognition site for Clp subunits near the GluTR N-terminus is masked (*Apitz et al., 2016*). Moreover, GluTR1 is tightly controlled to avoid unwanted tetrapyrrole synthesis, for example, during darkness, but is needed for high ALA synthesis rates in light-grown tissue. Hence, an effector molecule is needed to regulate the conditional stability of GluTR1. We propose that heme functions as the trigger for reduced ALA synthesis by binding to GBP. This assumption is supported by heme-dependent attenuation of the GluTR1-GBP interaction (*Figure 7A,B*) and the strongly decreased affinity of heme-bound GBP for GluTR1 demonstrated by MST measurements (*Figure 7C,D*). The negative impact of heme on the GBP-GluTR interaction is in agreement with decreased stimulation of GluTR activity by GBP in the presence of heme (*Zhao et al., 2014*). GBP binds to both *Arabidopsis* GluTR isoforms, encoded by *HEMA1* and *HEMA2* (*Czarnecki et al., 2011*), suggesting that GluTR2 is also regulated by heme-bound GBP. However, GluTR2 is not degraded during ALA feeding in our study (*Figure 1A*). A sequence alignment of *Arabidopsis* GluTR1 and GluTR2 reveals that the N-terminal region of GluTR2 lacks four amino acids relative to the RED of GluTR1 (*Figure 8—figure supplement 1*). Whether this difference between the two GluTR isoforms accounts for their different susceptibilities to heme-dependent breakdown by the Clp protease remains to be determined.

It is comprehensible that the stability of GluTR is ensured by GBP-dependent stabilization, but instantaneous inhibition of ALA synthesis is predominantly mediated by FLU, which inactivates GluTR in the light and in darkness (*Hou et al., 2019*; *Meskauskiene and Apel, 2002*; *Meskauskiene et al., 2001*). Although binding to FLU likely prevents Clp-dependent degradation of GluTR (*Hou et al., 2019*), we propose that FLU and heme act in concert to control ALA synthesis in plants (*Goslings et al., 2004*).

## Implications for the understanding of TBS pathway regulation

Photosynthetic cells in plants thus rely on at least two mechanisms to rapidly respond to changing needs for tetrapyrroles (FLU and heme). In contrast, heterotrophic tissues, which do not synthesize Chl (e.g. roots), do not utilize the FLU-POR-Pchlide inactivation complex to control ALA synthesis (*Kauss et al., 2012*). Here, heme presumably serves as the sole signal that triggers the adjustment of GluTR amounts and ALA synthesis rates. Although the precise mechanism is still unknown, the heme status also controls GluTR amounts and activity in bacteria (*Wang et al., 1999a*; *Wang et al., 1999b*; *de Armas-Ricard et al., 2011*), implying that the GluTR proteolysis mechanism evolved early in evolutionary history. Phylogenetic analysis reveals, however, that homologs of GBP and the RED of GluTR1 are only found in land plants (*Figure 8—figure supplement 2*). In light of the fact that

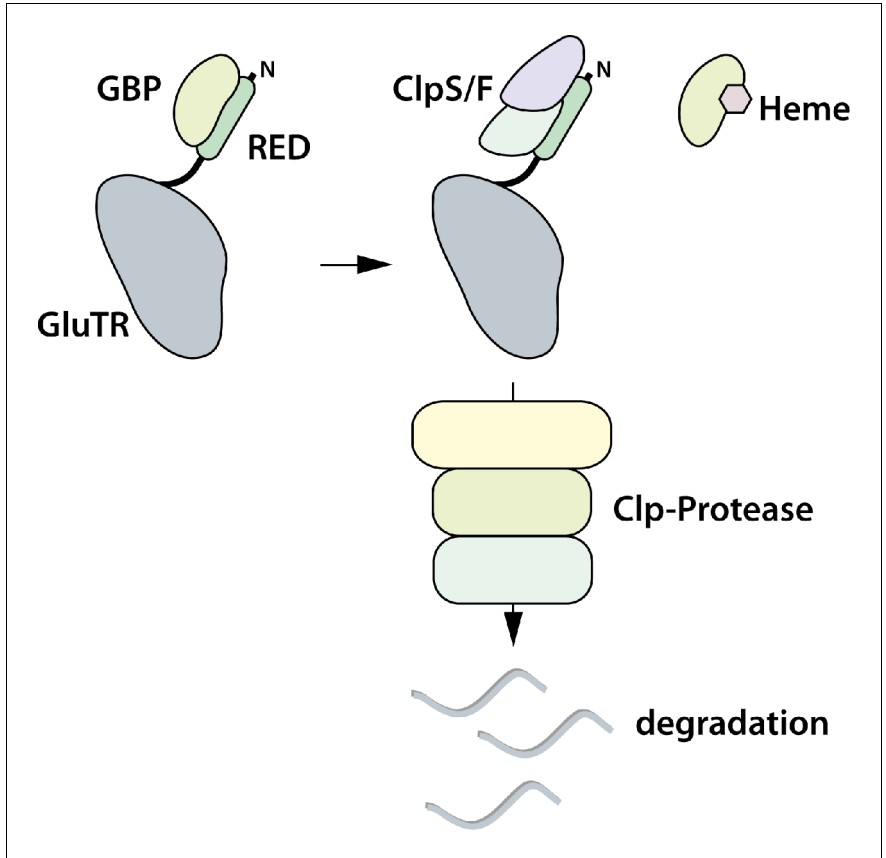

**Figure 8.** Proposed model for heme-dependent GluTR1 degradation by Clp protease. In the absence of heme, GBP binds to the 'regulatory domain' (RED, formerly 'heme binding domain') and prevents degradation of GluTR1. When heme levels increase, release of GBP from GluTR1 enables the binding of Clp components to the latter and its concomitant proteolytic degradation.

DOI: https://doi.org/10.7554/eLife.46300.012

The following figure supplements are available for figure 8:

**Figure supplement 1.** Alignment of the amino-acid sequences of *Arabidopsis* GluTR1 (encoded by *HEMA1*), GluTR2 (*HEMA2*) and the regulatory domain of GluTR1 (RED, formerly heme-binding domain - HBD).

DOI: https://doi.org/10.7554/eLife.46300.013

**Figure supplement 2.** Distribution of GBP and regulatory domain (RED, formerly heme-binding domain - HBD)-containing GluTR isoforms in photosynthetic organisms.

DOI: https://doi.org/10.7554/eLife.46300.014

angiosperms lacking dark-dependent POR (DPOR) need to control ALA synthesis during light-dark cycles tightly, it is reasonable to assume that GBP and RED have coevolved for the regulation of GluTR stability by heme. In this line, regulation of Chl synthesis by modulated GluTR stability was proposed to play a minor role in Chlamydomonas (*Nogaj et al., 2005*) suggesting a posttranslationally adjusted GluTR activity.

The presented model also explains phenomena and experimental results published in the past. A *flu* phenotype has been noted in the *fc2-2* mutant, that is, light-dependent necrosis owing to the overaccumulation of TBS intermediates during darkness (*Scharfenberg et al., 2015*; *Woodson et al., 2015*). This observation can now be explained by the diminished synthesis of heme functioning as the stimulus for GluTR1 degradation (*Figure 5*). Furthermore, the *aurea* (*au*) and *yellow-green 2* (*yg-2*) mutants of tomato, which are defective in the phytochromobilin synthase and heme oxygenase (HO/HY1) respectively, show a pale-green phenotype which is related to enhanced inhibition of ALA synthesis (*Terry and Kendrick, 1996*; *Terry and Kendrick, 1999*). Accumulation of heme owing to impaired heme conversion to phytochromobilin in these mutants was assumed to

inhibit GluTR activity. This was also observed in assays with a recombinant plant GluTR (*Vothknecht et al., 1998*). Additionally, introduction of a *HO1* mutation into a *flu* mutant background rescued the *flu* phenotype by alleviating Pchlide accumulation (*Goslings et al., 2004*). Furthermore, early work on ALA synthesis in cucumber plastids and etiolated bean seedlings treated with dipyridyl and heme, respectively, revealed heme-dependent inhibition of ALA synthesis (*Chereskin and Castelfranco, 1982*; *Duggan and Gassman, 1974*). Based on our new findings, it is reasonable to propose that diminishes ALA synthesis in *au*, *yg-2* and *flu/hy1* mutants results from GluTR breakdown which is due to alterations of heme metabolism and modified levels of regulatory heme.

## Materials and methods

### Plant material and growth conditions

The *Arabidopsis* genotypes used in this study included the wild type (Col-0), *clpc1-1* (SALK_014058), *clpr2-1* (SALK_046378, *Kim et al., 2009*), *fc1-1* (SALK_150001), *fc2-2* (SAIL_20_C06, *Scharfenberg et al., 2015*), *hema1/pHEMA1::HEMA1*(WT) and *hema1/pHEMA1::HEMA1*(ΔRED) (*hema1* background: SALK_053036, *Apitz et al., 2016*) as well as *gun5-2* (SALK_062726, *Huang and Li, 2009*). PCR-based genotyping was performed on genomic DNA using previously published primers. With the exception of *gun5-2*, plants were grown on soil for 14–21 days in continuous light (100 µmol photons $m^{-2}$ $s^{-1}$) Seeds of heterozygous *gun5-2* were surface-sterilized using ethanol and plated on Murashige and Skoog medium (4.4 g L21 Murashige and Skoog, 0.05% [w/v] MES, 1% [w/v] sucrose, 0.8% [w/v] agar, pH 5.7). White homozygous *gun5-2* seedlings were harvested after 4 weeks of growth in continuous light (100 µmol photons $m^{-2}$ $s^{-1}$).

### Experimental design and statistical analysis

If not otherwise stated, at least three replicates for each genotype and/or treatment were harvested and processed individually. Every harvested sample (one biological replicate) consisted of leaves from three-to-five 14-day-old plants, except in the experiments with the strongly growth retarded white *gun5-2* mutants, in which at least ten individuals were pooled per sample. All experiments were performed at least twice, and one representative experiment is shown. Statistical analysis was performed before normalization of the data by means of Student's t-test, and significant difference was assumed with $p<0.05$ (n=$\geq$3). Data are presented as mean $\pm$ standard deviation (sd).

### 5-Aminolevulinic acid treatment

ALA-feeding was performed by incubating leaves in buffer (20 mM TRIS, pH 7.4) or in buffer supplemented with 1 mM ALA. Samples were harvested at the indicated time points, dried on a paper towel and frozen in liquid nitrogen. Samples were stored at −80°C until further use.

### Growth and treatment of photoautotrophic *Arabidopsis* cell culture

A photoautotrophic *Arabidopsis* (Col-0) cell culture (PA) (*Hampp et al., 2012*) was used in this study. The PA cultures were maintained in two-tiered flasks placed in an orbital shaker under continuous illumination. The lower compartment contained a carbonate buffer that maintained a level of approximately 2% $CO_2$ in the upper compartment. Cultures were routinely diluted with growth medium (3.6 g/l Gamborg B5 (Duchefa), 1 g/l 2,4 dichloro-phenoxy-acetic acid, pH 5.7) after 3 weeks of growth. Experiments were performed with cultures grown for 7 days after the dilution. For the experiments, PA cells were harvested by centrifugation (8000 g, RT) and transferred to growth medium either without addition of further chemicals or supplemented with 500 µM 2,2'-DP or 100 µM ACI. After pre-incubation with ACI or DP for 2 hr in darkness, the buffer was exchanged for media supplemented with either 1 mM ALA, ACI + ALA, DP + ALA or buffer without additives. The treatment was performed for 16 hr in the dark. Cell culture samples (each 1 ml) were harvested by centrifugation (14,000 g, 4°C) and frozen in liquid nitrogen.

### Protein extraction and western-blot procedures

Total leaf protein was extracted from ground leaf material using extraction buffer (2% (w/v) SDS, 56 mM NaCO₃, 12% (w/v) sucrose, 56 mM DTT and 2 mM EDTA, pH 8.0). Homogenized samples were

heated for 20 min at 70°C, centrifuged for 10 min (16,000 g, RT) and the supernatant was transferred to a new reaction tube. Protein concentration was determined using the BCA protein assay kit (ThermoFischer Scientific, MA). Samples (10–15 µg of protein) were fractionated by electrophoresis on SDS-polyacrylamide gels (12%) and blotted onto nitrocellulose membranes (Amersham Protran, GE Healthcare, UK). Membranes were probed with protein-specific antibodies, according to *Sambrook and Russell (2001)*.

## RNA extraction and qPCR analysis

For qPCR analysis leaf material was harvested, frozen in liquid nitrogen and ground in a steel-ball mill. RNA was isolated using the citric acid method (*Oñate-Sánchez and Vicente-Carbajosa, 2008*). After DNase I treatment (ThermoFisher Scientific, MA) aliquots (1–2 µg) of RNA were transcribed into cDNA using Moloney Murine Leukemia Virus (M-MuLV) reverse transcriptase (ThermoFisher Scientific) and oligo dT(18) primer. All steps were performed essentially according to the manufacturer's protocol. The qPCR analysis was carried out in a CFX96-C1000 96-well plate thermocycler (Bio-Rad, CA) using SYBR green dye (Bio-Tool). The primers used are listed in *Supplementary file 1*. Calculation of gene expression levels was performed with the Bio-Rad CFX-Manager Software 1.6 using the $\Delta\Delta C(t)$ method and *ACTIN 2.2* as reference.

## Analysis of intermediates and end-products of tetrapyrrole biosynthesis

End-products and intermediates of the TBS pathway were extracted from ground leaves/PA cell cultures after lyophilization of the sample material. The dried powder was homogenized in 1 ml of acetone: 0.2 M $NH_4OH$ (9:1), incubated for 1 hr at $-20°C$ and centrifuged for 30 min (14,000 g, 4°C). The supernatant, containing (Mg) porphyrins and Chls, was analyzed by HPLC (see below). Non-covalently bound (ncb) heme was extracted from the remaining pellet. After resuspension of the pellet in 200 µl AHD (acetone:concentrated HCl:DMSO, 10:0.5:2), samples were incubated for 15 min at room temperature (RT). After centrifugation (16,000 g, RT), the supernatant was analyzed by HPLC. HPLC analysis was performed on Agilent LC systems following the methods described in *Supplementary file 2* and using authentic standards for the purpose of peak quantification.

## Determination of free heme

Free heme was determined using the method published by *Espinas et al. (2012)*. Briefly, 4–8 mg of ground and lyophilized leaf material was mixed with 1 ml of 100% acetone, incubated for 5 min at RT and centrifuged for 10 min at 4°C (16,000 g). Twenty microliters of the supernatant was combined with 80 µl of apo-horseradish peroxidase (HRP) mixture (250 mM TRIS, pH 8.4) containing 231 nM apo-HRP 4C (EC 1.11.1.7, BBI solutions, Crumlin, UK). After reconstitution of the peroxidase with the extracted heme for 30 min at RT, 100 µl of western blot reagent (Clarity, Bio-Rad, CA) was added. Subsequently, the sample was transferred to a black microtiter plate, and the chemiluminescent signal was recorded for between 30 s and 2 min using a CCD camera (ECL chemostar, INTAS, Germany).

## Chloroplast isolation and treatment

Chloroplasts were isolated from 4-week-old *Arabidopsis* Col-0 plants grown on soil under short-day conditions, following previously published protocols (*Richter et al., 2013*). The chloroplasts were resuspended in resuspension buffer (0.3 M sorbitol, 20 mM Tricine-KOH, 2.5 mM EDTA, 5 mM $MgCl_2$, pH 8.4). Aliquots (100 µl) of chloroplast suspension were mixed with porphyrin solution to obtain a final concentration of 5 µM hemin, ProtoIX, MgP and MgPMME (stocks 500 µM DMSO) or 1 µM Pchlide. 1% (v/v) DMSO was used as control. After dark incubation with gentle agitation for the indicated times aliquots, of the chloroplasts were mixed with protein loading buffer (62.5 mM Tris, 10% glycerol [v/v], 2% SDS [w/v], 100 mM DTT, 0,05% bromophenol blue [w/v], pH 6.8) and analyzed by western blotting.

## Expression and purification of proteins

GluTR1 and GBP were expressed and purified from the *E. coli* BL21 strain as described previously (*Apitz et al., 2016*; *Czarnecki et al., 2011*; *Wang et al., 2018*). 6xHIS-tagged proteins were purified using Ni-NTA (Thermo Scientific) and glutathione-S-transferase (GST)-tagged proteins were purified

using Glutathione Sepharose (GS) 4B (GE Healthcare). Extraction and purification was performed following manufacturers' protocols. After dialyzing the proteins into PBS (137 mM NaCl, 2.7 mM KCl, 10 mM $Na_2HPO_4$, 1.8 mM $KH_2PO_4$), glycerol was added to a final concentration of 10% (v/v), and protein aliquots were stored at $-80°C$.

## Heme binding assay

Hemin agarose was prepared according to *Busch et al. (2017)*. Briefly, 2 ml of aminoethyl-agarose beads (aminoethyl 6% agarose beads cross-linked 50 % v/v, Goldbio, MO) were transferred to a column and washed with 7 ml water, 7 ml 33% dimethylformamide (DMF), 7 ml 66% DMF and finally with 14 ml of 100% DMF. A 50 mM hemin solution (in DMF) was mixed with an equal volume of 1,1'-carbonyldiimidazole (CDI, 26.6 mg/ml DMF, Sigma-Aldrich, Munich, Germany) and incubated for 15 min at 80°C (dark). To remove undissolved hemin, the hemin:CDI mixture was centrifuged for 30 min at RT (16,000 g). The supernatant was mixed with the DMF-washed agarose beads and incubated for 18 hr on a rotary incubator in the dark. The next day, the matrix was washed with approximately 70 column volumes (CV) of 25% pyridine (Sigma-Aldrich, Munich, Ger) until the flow-through was transparent. After washing with 10 CV of double-distilled water, the matrix was equilibrated with 10 CV PBS. Finally, the matrix was resuspended in PBS, resulting in 33% hemin-agarose solution. Aliquots of the matrix were stored at 4°C until further use. To assay for hemin binding, 10 µg of recombinant proteins were incubated with 75 µl of the hemin-agarose or PBS equilibrated agarose beads (control) in a final volume of 500 µl PBSplus (PBS supplemented with 10% glycerol, 0.2% Tween 20). After 30 min of incubation on a rotary incubator at RT, samples were transferred to centrifuge columns and centrifuged at 1500 rpm (RT) for 1 min. Columns were washed 10 times with PBSplus supplemented with 500 mM NaCl. Bound proteins were eluted using 30 µl of PAGE loading buffer and were analyzed by SDS-PAGE.

## Competitive binding assays

For (competitive) pull-down experiments 10 µg of GST-GBP were mixed with 10 µl of GSH-Sepharose (equilibrated with PBSplus, see above) in a final volume of 500 µl PBSplus. Hemin was added to a final concentration of 500 µM. After 30 min of incubation on a rotary incubator (RT) 10 µg of HIS-GluTR1 was added and incubated again for 30 min at RT. The assay mixtures were centrifuged for 1 min at 215 g (RT), and the supernatant was discarded. The pelleted matrix was washed ten times with PBSplus supplemented with 500 mM NaCl. Proteins were eluted using 30 µl SDS-PAGE loading buffer. Because GST-GBP and 6xHIS-GluTR1 are of almost the same molecular weight, western blotting and the 6xHIS-tag specific antibody were used to detect the presence of 6xHIS-GluTR1 in the eluates.

All MST reagents and consumables were purchased from NanoTemper (Munich, Germany). The purified HIS-GluTR and HIS-GBP were diluted in PBS to a final concentration of 20 µM. The proteins were labeled with the RED-NHS (Amine Reactive) protein labeling kit (second generation) according to the protocols provided. Proteins were further diluted with PBS (f.c. 400 nM) after labelling. The binding reactions consist of 200 nM labeled protein (constant) and decreasing amounts of the binding partner (starting from 7.5 µM). MST measurements were performed in a Monolith NT.115 device using premium coated capillaries. Excitation and MST power were set to 100% and 40%, respectively. In the case of labeled HIS-GBP, the initial fluorescence quenching was used to calculate the Kd value. For the experiments with labeled HIS-GluTR, the MST signal observed 1.5 s after turning on the laser was used instead.

## Additional information

### Funding

| Funder | Grant reference number | Author |
|---|---|---|
| Deutsche Forschungsgemeinschaft | 317556048 | Bernhard Grimm |

The funders had no role in study design, data collection and interpretation, or the decision to submit the work for publication.

## Author contributions
Andreas S Richter, Conceptualization, Data curation, Formal analysis, Supervision, Validation, Investigation, Visualization, Methodology, Writing—original draft, Project administration, Writing—review and editing; Claudia Banse, Formal analysis, Investigation; Bernhard Grimm, Resources, Supervision, Funding acquisition, Writing—original draft, Writing—review and editing

## Author ORCIDs
Andreas S Richter ⓘ https://orcid.org/0000-0002-2293-7297
Bernhard Grimm ⓘ https://orcid.org/0000-0002-9730-1074

## Decision letter and Author response
Decision letter https://doi.org/10.7554/eLife.46300.019
Author response https://doi.org/10.7554/eLife.46300.020

# Additional files

## Supplementary files
• Supplementary file 1. HPLC methods.
DOI: https://doi.org/10.7554/eLife.46300.015
• Supplementary file 2. qPCR primers used in this study.
DOI: https://doi.org/10.7554/eLife.46300.016
• Transparent reporting form
DOI: https://doi.org/10.7554/eLife.46300.017

## Data availability
All data generated or analysed during this study are included in the manuscript and supporting files.

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
