## [Decision Letter]

Thank you for submitting your article "The GluTR-binding protein is the heme-binding factor for feed-back control of glutamyl-tRNA reductase" for consideration by *eLife*. Your article has been reviewed by three peer reviewers, and the evaluation has been overseen by a Reviewing Editor and Michael Marletta as the Senior Editor. The following individuals involved in review of your submission have agreed to reveal their identity: Mats Hansson (Reviewer #3).

The reviewers have discussed the reviews with one another and the Reviewing Editor has drafted this decision to help you prepare a revised submission.

The manuscript by Richter et al. presents data in support of a role for heme in controlling ALA production in plants via regulation Clp proteolysis of GluTR. The basic model proposes that GBP normally binds to an amino-terminal extension of GluTR thereby preventing Clp-mediated proteolysis of GluTR but when cellular heme is in excess, the heme binds to GBP causing it to release from GluTR and allow degradation of GluTR to occur. The data presented in this manuscript is consistent with this model. In general, the paper is technically sound. But reviewers have raised important concerns which need to be addressed prior to a final decision. Major concerns are (a) the novelty of the findings in light of published work from the same group (example: Apitz et al.,2014 and 2016); (b) presentation style which will require proof-reading and careful editing; and (c) calculating statistical significance. I have summarized the three reviewers' comments below.

Essential revision requirements:

1) As mentioned throughout the manuscript, the report here builds on your previous work. In particular the Apitz et al. paper, while not containing replicated data, does contain much of the information presented herein. The clear novel finding in the current work is the demonstration that heme, which was previously shown to bind to GBP, does not bind to GluTR. Thus, you suggest renaming the "heme binding domain" of GluTR as the "regulatory domain" since it is GBP-binding domain, not a heme binding domain. In the end, we conclude the central finding is important enough advance; however, some text revision focusing more attention on this finding and less on recounting past results in Apitz et al. is required.

2) Another shortcoming is the written presentation. There is very considerable repetition between the Introduction, Results and Discussion. There is too much discussion in the Results section and the Discussion is overly long. The first paragraph of subsection “Implications for the understanding of TBS pathway regulation” is superfluous to the aim of the manuscript since by that point in the manuscript the model has been presented and discussed multiple times. Any resubmission of the current offering should be reduced in length by at least a third.

3) The first two sentences of the Abstract are not completely accurate: whether "ALA synthesis is rate-limiting in *all* organisms" is overly broad and lacks proof.

4) A major concern is that the authors should go through the figures and think of the statistical values, i.e. which numbers are actually significant in general and how should standard deviation be calculated on ratios which are presented?

---

## [Author Response]

Essential revision requirements:1) As mentioned throughout the manuscript, the report here builds on your previous work. In particular the Apitz et al. paper, while not containing replicated data, does contain much of the information presented herein. The clear novel finding in the current work is the demonstration that heme, which was previously shown to bind to GBP, does not bind to GluTR. Thus, you suggest renaming the "heme binding domain" of GluTR as the "regulatory domain" since it is GBP-binding domain, not a heme binding domain. In the end, we conclude the central finding is important enough advance; however, some text revision focusing more attention on this finding and less on recounting past results in Apitz et al. is required.2) Another shortcoming is the written presentation. There is very considerable repetition between the Introduction, Results and Discussion. There is too much discussion in the Results section and the Discussion is overly long.

What follows is the response to points 1 and 2.

When statements and previously published results were reiterated it was done to improve readability, to provide the theoretical background for the results and to facilitate the understanding of conclusions. Because the previous work builds the framework for the presented experiments, the new results are intentionally discussed with respect to the previous findings. It is important to note that, although already presented before, the Clp-GBP-dependent regulation of GluTR stability is better understood in the view of our new findings, when heme as an important regulator is included in the model. However, we reduced the amount of repetitions wherever possible, moved or deleted discussions from the Results section and carefully revised the Discussion to emphasize the novelty of our work.

The first paragraph of subsection “Implications for the understanding of TBS pathway regulation” is superfluous to the aim of the manuscript since by that point in the manuscript the model has been presented and discussed multiple times. Any resubmission of the current offering should be reduced in length by at least a third.

Despite the recommendation, we feel that the additional explanations provided in this paragraph provide a broader context and discuss the importance of the regulatory mechanism for other cell type (e.g. in heterotrophic tissue), and overall in organisms. However, following your recommendation, we revised this paragraph. By reducing the length of the entire main text to 4,200 words, we hope to meet the requirements of the journal.

3) The first two sentences of the Abstract are not completely accurate: whether "ALA synthesis is rate-limiting in all organisms" is overly broad and lacks proof.

We have specified the corresponding sentences.

4) A major concern is that the authors should go through the figures and think of the statistical values, i.e. which numbers are actually significant in general and how should standard deviation be calculated on ratios which are presented?

Statistical analysis was added to the figures and explanations are provided in the figure legends.